# Accuracy of efficient data methods to determine the incidence of hospital-acquired thrombosis and major bleeding in medical and surgical inpatients: a multicentre observational cohort study in four UK hospitals

Daniel Horner ,[1,2] Saleema Rex,[3] Charles Reynard ,[4] Matthew Bursnall,[3] Mike Bradburn ,[3] Kerstin de Wit ,[5,6] Steve Goodacre ,[7] Beverley J Hunt[8]

For 'Presented at statement' see end of article.

For numbered affiliations see end of article.

**Correspondence to**
Daniel Horner;
daniel.horner@nca.nhs.uk

## ABSTRACT

**Objectives** We evaluated the accuracy of using routine health service data to identify hospital-acquired thrombosis (HAT) and major bleeding events (MBE) compared with a reference standard of case note review.

**Design** A multicentre observational cohort study.

**Setting** Four acute hospitals in the UK.

**Participants** A consecutive unselective cohort of general medical and surgical patients requiring hospitalisation for a period of >24 hours during the calendar year 2021. We excluded paediatric, obstetric and critical care patients due to differential risk profiles.

**Interventions** We compared preidentified sources of routinely collected information (using hospital coding data and local contractually mandated thrombosis datasets) to data extracted from case notes using a predesigned workflow methodology.

**Primary and secondary outcome measures** We defined HAT as objectively confirmed venous thromboembolism occurring during hospital stay or within 90 days of discharge and MBE as per international consensus.

**Results** We were able to source all necessary routinely collected outcome data for 87% of 2008 case episodes reviewed. The sensitivity of hospital coding data (International Classification of Diseases 10th Revision, ICD-10) for the diagnosis of HAT and MBE was 62% (95% CI 54 to 69) and 38% (95% CI 27 to 50), respectively. Sensitivity improved to 81% (95% CI 75 to 87) when using local thrombosis data sets.

**Conclusions** Using routinely collected data appeared to miss a substantial proportion of outcome events, when compared with case note review. Our study suggests that currently available routine data collection methods in the UK are inadequate to support efficient study designs in venous thromboembolism research.

**Trial registration number** NIHR127454.

## STRENGTHS AND LIMITATIONS OF THIS STUDY

⇒ This study used predefined outcomes and international consensus definitions to evaluate the accuracy of routinely collected data for identification of hospital-acquired thrombosis and major bleeding events, during hospital admission.

⇒ All data abstractors were blinded to routine data sources, limiting bias in case ascertainment.

⇒ Research assistants varied in clinical experience by site, which may introduce variability in outcome reporting.

⇒ Our findings may lack generalisability to other healthcare settings, given the UK context.

attributable morbidity and mortality.[1] At least half of all VTE occurs during hospitalisation, or up to 90 days following discharge; such cases are described as hospital-acquired thrombosis (HAT).[2] Many of these events are potentially preventable through patient education and provision of thromboprophylaxis to those at risk.

Research into thromboprophylaxis often requires large sample sizes to identify small but important differences in clinically relevant events, such as HAT and/or major bleeding. Study protocols will often necessitate examination of case notes to identify outcome events, which can be time consuming and expensive. This is particularly relevant for external validation of new clinical decision rules or risk assessment models (RAMs) which aim to guide prescribing of thromboprophylaxis for hospital inpatients.[3–5]

Using routine health service data to identify outcome events could markedly improve the efficiency of research and facilitate studies

## BACKGROUND

Venous thromboembolism (VTE) remains a major global health burden, with significant

with large sample sizes at acceptable cost. However, this approach requires confirmatory evidence that routine data sources accurately identify outcome events.

Several mechanisms already exist for routine identification of outcomes, including hospital coding, local VTE data sets, and pathology reporting (with thrombosis committee oversight). If such efficient methods could accurately ascertain relevant outcomes, large-scale studies would be theoretically deliverable.

We sought to evaluate the accuracy of using routine data to identify HAT and major bleeding events (MBE) compared with case note examination.

## METHODS

We conducted a multicentre observational cohort study within the context of a wider project examining the overall clinical and cost effectiveness of VTE RAMs.[6] The aim of this study was to estimate the accuracy and completeness of available coding data and local registry data to determine clinically relevant VTE and bleeding outcomes against case note review by trained research assistants.

We approached four National Health Service (NHS) sites to participate in this study; the Northern Care Alliance NHS Foundation Trust in Salford, Manchester University Hospitals NHS Foundation Trust, the Northern General Hospital in Sheffield and Guy's & St Thomas' NHS Foundation Trust in London.

### Study population

We identified a consecutive, unselected cohort of general medical and surgical patients requiring hospital admission at each site during the calendar year 1 January to 31 December 2019. We chose 2019 because of concern that patients admitted during the subsequent COVID-19 pandemic might represent an atypical cohort. We collated data on all risk assessments that clinical staff performed prospectively at the point of hospital admission, then scrambled episodes into randomly assorted batches of 50 (referred to as 'A' batches) to ensure diversity in specialty presentation and mitigate seasonal bias. We collated cases (or records) in batches of 50 to facilitate iterative and incremental case ascertainment; initial hospital downloads were often in excess of 50 000 case episodes. In order to keep workflow manageable and organised, we worked through batches of 50 records at a time and reported routinely to a steering committee, who provided guidance on study delivery. We excluded paediatric patients (age <16), anyone requiring critical care admission (defined as level 2 care or above) and pregnant/postpartum patients due to differential VTE and bleeding risks, as outlined in the wider study protocol.[7]

### Study design

For each patient episode, we extracted baseline demographics and prospectively collected data on VTE risk assessment (where available) from the electronic health

record (EHR), with support from business intelligence teams. Risk assessments were captured differently at each site, including the use of a paper proforma, dichotomous output on electronic prescribing (low/high risk) or through a detailed structured note within the electronic healthcare record. Example images/screenshots for each site can be found in the online supplemental material. All four sites used the Department of Health tool to facilitate VTE risk assessment; this RAM has been developed by expert consensus and is recommended in national UK guidance.[8] Recent survey data suggest that the tool is used by >80% of NHS sites, despite limited available data on external validation.[9] This tool confers a high rate of prescribing in comparison with other RAMs, as highlighted in a recent practice review.[3]

Research assistants at each site undertook retrospective case note review for each patient episode through shared primary and secondary care EHR. We utilised EHR to access primary care data on hospital attendance, diagnoses and investigation within the relevant time periods. Primary care EHR systems varied by trust. We used secondary care EHR to identify hospital reattendance, investigations, diagnostic imaging and confirmed diagnoses (via discharge summary or note entry). Secondary care EHR systems varied by trust, but access to radiology investigations was universal within the patient archiving and communication system. We extracted descriptive data on relevant clinical outcomes such as the subsequent diagnosis of VTE/HAT, major bleeding and clinically relevant non-MBEs as per internationally agreed definitions.[10 11] We defined VTE as any pulmonary embolism (PE) or deep vein thrombosis (DVT) identified in routine care by the treating clinical team, in accordance with International Society on Thrombosis and Haemostasis (ISTH) common data elements.[12] Superficial venous thrombosis was specifically excluded from this definition. We defined HAT in accordance with the definition proposed by NHS England (any new episode of VTE occurring during hospitalisation or within 90 days of discharge, following an inpatient stay of ≥2 days or a surgical procedure under general/regional anaesthesia).[13] Data extractors were trained in identification of these outcomes and followed a detailed workflow diagram (online supplemental material). Data abstractors were blinded to batch allocation, final International Classification of Diseases 10th Revision (ICD-10) coding, local database entries and the final analysis plan.

Following case note review, we collected data elements from multiple data sources for each patient episode to evaluate their combined accuracy against case note review outcomes. Data sources for interrogation were identified a-priori and included the following; ICD-10 diagnostic codes judged relevant to thrombosis or bleeding by the project management group (a-priori, shown in the online supplemental material); Emergency Care Data Set/ Systematised Nomenclature of Medicine(SNOMED) clinical terms or codes relevant to thrombosis or bleeding and contractual local HAT databases. HAT database

entries are co-ordinated by local thrombosis committees in accordance with NHS contract standards and include a contemporary register of all patients diagnosed with acute VTE at the hospital site, informed by radiology or identified by pathology at postmortem.[13] All cases are subsequently categorised by the local thrombosis committee as either de-novo VTE or HAT based on case review, local expert opinion and data on any preceding hospital admission (up to 90 days) or VTE diagnosis >24 hours following hospital admission. This database is maintained contemporaneously and provides an ongoing opportunity for hospitals to identify preventable HAT and conduct root cause analysis (RCA) for each episode, to promote learning and best practice. All data sources were interrogated for the duration of hospital stay and up to 90 days postdischarge, for each patient episode. Data sources were obtained through routine local business intelligence requests or direct approach to local coding teams. HAT database entries were obtained where feasible through local site thrombosis committee chairs.

Given the potential for negligible VTE/MBE in the wider study population (leading to limited information on the accuracy of efficient data methods), we augmented the overall sample with positive thrombosis and bleeding cases. We obtained positive cases through ICD-10 coding identification for VTE events (V batches), bleeding events (B batches) and positive VTE cases from local HAT database entries (H batches), identified as above and sourced from local thrombosis committee leads. Positive cases were batched and reviewed in accordance with the general study protocol. Data extractors were blinded to batch allocation.

## Outcomes

The following criteria were proposed to determine whether routine data identify outcome events with sufficient accuracy to support efficient methods:

1. Proportion of outcome events identified by routine data sources that are confirmed by record review (target 100%)
2. Proportion of cases with no outcome event identified by routine data sources that have an event identified on record review (target 0%).
3. Proportion of inpatients with data collected (target 90%).

## Statistical analysis

The accuracy of routinely recorded HAT and bleeding events was compared against direct case note review data for the cohort. Case note review determination of events was assumed to be the gold standard. Data are presented in contingency tables with sensitivity, specificity, positive and negative predictive values along with CIs calculated using the Wilson score method.[14]

The primary analysis included patients identified from all sources (A, V, B or H batches) with bleeding and HAT assumed to have not occurred unless coded as such in the relevant data, or detected following case note review.

In addition, two preplanned sensitivity analyses were undertaken:

1. Inclusion was limited to cases identified in routine case review (ie, 'A batch patients' only), with exclusion of all augmented sample cases.
2. Inclusion limited to participants for whom bleeding or HAT was definitively recorded

We took a conservative approach and interpreted missing or unknown endpoints as 'no event' with the exception of the second sensitivity analysis.

We originally planned to identify 3000 inpatients across four hospitals during a 12-month study period within the 2-year project plan, dependent on appointment of research assistants and time required for outcome ascertainment. This sample size was designed to allow key parameters to be estimated with a high degree of precision across the whole cohort (SE<1%). All sites failed to meet their sample target of 750 for reasons mainly related to the SARS-COV-2 pandemic, including redeployment of research staff to clinical care, delayed local approvals secondary to prioritised pandemic research and a longer than anticipated time for individual outcome ascertainment per case review

## Ethical aspects

The study received a favourable opinion from the Proportionate Review Subcommittee of the London—West London & GTAC Research Ethics Committee and approval from the HRA and Care Research Wales (HCRW) on 18 September 2019 (reference 19/LO/1303, IRAS project ID 262220).

Participating sites identified members of the clinical care team (research nurses or assistants predominately) to access patient records and extract clinical data using a predesigned and protected Microsoft Excel© database with embedded macro function, hosted at site. All data subsequently underwent local deidentification following completion and were exported to an independent team of statisticians at the Clinical Trials Research Unit (CTRU) in Sheffield, for collation and analysis.

All aspects of the data collection process, export, analysis and oversight were regularly reviewed by the internal Project Management Group including CTRU representation, and an external Trial Steering Committee (TSC), throughout the duration of the project.

We conducted this study in accordance with international Enhancing the QUAlity and Transparency Of health Research (EQUATOR) guidelines. A Strengthening the Reporting of Observational Studies in Epidemiology (STROBE) reporting checklist was used throughout to inform design, conduct and analysis of this observational cohort study and is included as online supplemental information.

## Patient and public involvement

Representatives of two Patient and Public Involvement (PPI) groups, thrombosis UK and Sheffield Emergency Care Forum (SECF) joined the research team and were

involved in developing the initial proposal and undertaking the wider study.

The SECF is a patient and public representative group with an interest in emergency care research. The forum has provided PPI for many emergency care research projects over then last 10 years.[15] Thrombosis UK is a charity that aims to identify, inform and partner the NHS, healthcare providers and individuals to work to improve prevention of VTE and the management and care of VTE events (see https://www.thrombosisuk.org/).

The PPI members were involved in determining the study design and ensuring that the proposal addressed the needs of patients and the NHS, while respecting the needs of potential participants. Their input regarding the importance of providing thromboprophylaxis for potential participants of any prospective cohort study and the need for such a study to yield reliable findings was instrumental in determining our approach to answering the research question. The PPI members also provided input at project management meetings and, where required, in day-to-day running of the project. The members used meetings and surveys of their wider PPI groups to enhance PPI in the project.

## RESULTS

We identified 2115 patients with an original hospital admission occurring in the calendar year 2019. Of these, 107 patient episodes were ineligible due to being pregnant or postpartum women (n=49); admitted to a critical care environment of level 2 or above (n=38); children aged under 16 (n=13) or for unrecorded reasons (n=7) leaving 2008 episodes for analysis. All episodes were suitable for data extraction and comparison to routine data sources.

Patient episodes showed an even balance of medical and surgical cases, but with a focus on emergency (73.7%) rather than elective (25.8%) admissions. A broad range of subspecialty interests were represented within the cohort. Median length of stay was 3 days (IQR 3 to 8) and mean length of stay 7.75 days (SD 16.5). Specialty groups with frequencies and cumulative percentages are shown in table 1. The vast majority of patient episodes (1809, 90.1%) were taken from 'A' batches. The total sample was augmented by 45 patients (2.2%) with potential bleeding events and 154 (7.7%) patients with potential VTE events. All sites contributed evenly to the sample with one exception; reduced numbers at this site reflect a delay to institutional approval during the pandemic, arising from a high burden of other clinical research studies and high staff turnover. Site and batch numbers are shown in table 2.

### Main findings

Contingency tables for the accuracy of routine data sources compared with case note review for both HAT and MBEs are shown in table 3. Sensitivity was 62% (95% CI, 54 to 69) for the use of ICD-10/SNOMED coding data to detect HAT events and 81% (95% CI, 75 to 87) for

local HAT database entries. Sensitivity by individual site ranged from 45% (95% CI, 28 to 63) to 72% (95% CI, 61 to 82) using ICD-10/SNOMED coding and 68% (95% CI, 51 to 84) to 94% (95% CI, 87 to 100) using local HAT database entries.

The sensitivity of ICD-10/SNOMED coding to detect MBEs identified by case note review was 38% (95% CI, 27 to 50). Sensitivity by individual site ranged from 22% (95% CI, 0 to 49) to 56% (95% CI, 37 to 75).

### Preplanned sensitivity analysis

A sensitivity analysis was conducted using only patient episodes obtained through 'A' batches, to remove augmentation of the sample and mitigate bias. The sensitivity of efficient data methods to detect key outcomes identified at case note review remained poor. These results are summarised in table 4.

We found the HAT event rate on case note review to be 29/1809 (1.6%, 95% CI, 1.0 to 2.2) and the MBE rate to be 45/1809 (2.5%, 95% CI, 1.8 to 3.2) within this large cohort of hospitalised patients receiving risk assessment and thromboprophylaxis in the context of routine care.

The proportion of outcome HAT events identified by routine data sources that were confirmed by record review (target 100%) was 71% (95% CI, 63 to 79) for ICD-10/SNOMED coding and 100% (95% CI, 97 to 100) for local HAT database entries. The proportion of cases with no HAT outcome event identified by routine data sources that had an event identified on record review (target 0%) was 3% (95% CI, 2 to 4) for ICD-10/SNOMED coding and 2% (95% CI, 1 to 2) for local HAT database entries. The proportion of MBEs identified by routine data sources that were confirmed by record review (target 100%) was 20% (95% CI, 13 to 27) for ICD-10/SNOMED coding. The proportion of cases with no major bleeding outcome event identified by routine data sources that have an event identified on record review (target 0%) was 2% (95% CI, 1 to 3) for ICD-10/SNOMED coding. We were able to collect outcome data for 1745/2008 (87%) inpatients (target 90%). This was <100% due to difficulty accessing the local HAT database at a single site. Excluding this issue, the other three sites all managed to collect relevant outcome data for at least 98% of patients.

## DISCUSSION

### Statement of principal findings

Our findings suggest that using currently available routine data for identification of HAT and MBE during hospital admission or within 90 days of discharge is not sufficiently sensitive to support a large data-enabled study. We failed to demonstrate feasibility for a number of predefined metrics and conclude that the use of routine data to identify outcomes would be highly likely to miss important events, and may erroneously identify false positive events.

**Table 1** Clinical category and admission type, with frequency and cumulative percentage

| | Frequency | Percentage | Cumulative |
|---|---|---|---|
| **Admission type** | | | |
| Missing | 1 | 0.05 | 0.05 |
| Elective | 518 | 25.80 | 25.85 |
| Emergency | 1480 | 73.71 | 99.55 |
| Unknown | 9 | 0.45 | 100 |
| Total | 2008 | 100 | |
| **Specialty group** | | | |
| Missing | 9 | 0.45 | 0.45 |
| Medical | 902 | 44.92 | 45.37 |
| Surgical | 951 | 47.36 | 92.73 |
| Tertiary specialty | 146 | 7.27 | 100 |
| Total | 2008 | 100 | |
| **Clinical category** | | | |
| Missing | 9 | 0.45 | 0.45 |
| Acute medicine | 340 | 16.93 | 17.38 |
| Ageing and complex medicine | 133 | 6.62 | 24.0 |
| Cardiology | 41 | 2.04 | 26.04 |
| Cardiothoracic surgery | 87 | 4.33 | 30.38 |
| Dermatology | 2 | 0.1 | 30.48 |
| Emergency medicine | 87 | 4.33 | 34.81 |
| Gastroenterology | 61 | 3.04 | 37.85 |
| General surgery | 285 | 14.19 | 52.04 |
| Medical: other | 169 | 8.42 | 60.46 |
| Neurology | 10 | 0.5 | 60.96 |
| Neurorehabilitation | 2 | 0.1 | 61.06 |
| Neurosurgery | 39 | 1.94 | 63.0 |
| Gynaecology | 57 | 2.84 | 65.84 |
| Renal medicine | 26 | 1.29 | 67.13 |
| Respiratory | 63 | 3.14 | 70.27 |
| Rheumatology | 2 | 0.1 | 70.37 |
| Trauma and orthopaedics | 158 | 7.87 | 78.24 |
| Upper GI surgery | 13 | 0.65 | 78.89 |
| Urology | 107 | 5.33 | 84.22 |
| Surgery: other | 170 | 8.47 | 92.69 |
| Tertiary specialty: other | 147 | 7.32 | 100 |
| Total | 2008 | 100 | |

GI, gastrointestinal.

### Strengths and weaknesses of the study

We engaged a combination of digitally mature and paper-based UK NHS sites in this study, used strict consensus definitions for VTE/bleeding events and evaluated only predefined efficient data sources. We also used topic experts and research staff to iteratively develop our data collection tool and workflow diagram, to limit subjective interpretation of case note data. However, there are limitations to this work. We evaluated patient episodes from large urban hospital sites, two of which are VTE exemplar centres and three of which are tertiary centres, which may limit external validity. Research assistants across sites varied in seniority and clinical experience; although all sites had a principal investigator and strict working definitions for outcome events, this may have introduced variation in reporting. We did not achieve our intended target of 3000 patients. However, it is important to note that the overall results within our cohort of 2008 patients are well

**Table 2** Number of cases submitted by site and batch type

| | Batch | | | | |
|---|---|---|---|---|---|
| | A | B | H | V | Total |
| London | 504 | 0 | 21 | 0 | 525 |
| Manchester | 241 | 0 | 0 | 0 | 241 |
| Salford | 570 | 45 | 44 | 46 | 705 |
| Sheffield | 494 | 0 | 43 | 0 | 537 |
| Total | 1809 | 45 | 108 | 46 | 2008 |

A, patient admissions requiring routine risk assessment; B, potential cases of bleeding (selected from relevant ICD-10 codes); H, cases of hospital-acquired thrombosis (HAT) identified through local thrombosis committee infrastructure; V, potential cases of venous thromboembolic disease (selected from relevant ICD-10 codes).
ICD-10, International Classification of Diseases 10th Revision.

outside of feasibility targets and sensitivity values were universally poor. We do not envisage that adding further cases would have significantly affected these values. Finally, we did not routinely collect individual patient characteristics, so do not report HAT or MBE stratified by relevant variables (such as the use of thromboprophylaxis).

## Strengths and weaknesses in relation to other studies, discussing important differences in results

Previous international work in this area is conflicting. A comparison of hospital episode statistics (HES) data to general practice records in England reported in 2012, initially concluded reliable identification of vascular disease (derived from ICD-10 coding data).[16] However, this analysis was restricted to PE from a VTE perspective and sought only to correlate disease states, rather than identify new case episodes. Several authors have used primary care research data sets correlated to evidence of anticoagulation or other secondary care data to identify VTE events, with reported reliable capture. This work does not seek to discriminate between index presentation of VTE and downstream development of HAT.[17 18]

A systematic review, with searches run in July 2010 and published in 2012, summarised findings on this topic from 19 studies. The positive predictive value (PPV) for PE ICD-10 codes ranged from 24% to 92%, with higher values from certain combinations of codes. PPV values for DVT codes ranged from 31% to 97%. More recently, a cross-sectional North American study compared ICD-10 codes for VTE in hospitalised medical patients to a 'gold standard' manual review of clinical data in 4000 patients.[19] The authors report a sensitivity of 63% for any DVT and a sensitivity of 83% for PE, implying further discrepancy

**Table 3** Contingency tables for main outcomes

| | | HAT from case note review | | |
|---|---|---|---|---|
| | | Yes | No | |
| HAT from ICD-10/ SNOMED codes | Yes | 95 | 39 | 71% (63%, 79%) True positive rate and 95% CI |
| | No | 59 | 1815 | 3% (2%, 4%) False negative rate and 95% CI |
| | | 62% (54%, 69%) Sensitivity and 95% CI | 98% (97%, 99%) Specificity and 95% CI | (n=2008) |
| | | Yes | No | |
| HAT from HAT RCA database | Yes | 122 | 0 | 100% (100%, 100%) True positive rate and 95% CI |
| | No | 29 | 1616 | 2% (1%, 2%) False negative rate and 95% CI |
| | | 81% (75%, 87%) Sensitivity and 95% CI | 100% (100%, 100%) Specificity and 95% CI | (n=1767)* |
| | | Major bleed from case note review | | |
| | | Yes | No | |
| Major bleed from ICD-10/SNOMED codes | Yes | 25 | 98 | 20% (13%, 27%) True positive rate and 95% CI |
| | No | 40 | 1845 | 2% (1%, 3%) False negative rate and 95% CI |
| | | 38% (27%, 50%) Sensitivity and 95% CI | 95% (94%, 96%) Specificity and 95% CI | (n=2008) |

*Manchester site excluded from this analysis as unable to access local HAT database.
HAT, hospital-acquired thrombosis; ICD-10, International Classification of Diseases 10th Revision; RCA, root cause analysis.

**Table 4** Sensitivity analysis using only A batch cases

| | | HAT from case note review | | |
|---|---|---|---|---|
| | | Yes | No | |
| HAT from ICD-10/ SNOMED codes | Yes | 18 | 18 | 50% (34%, 66%) True positive rate and 95% CI |
| | No | 11 | 1762 | 1% (0%, 1%) False negative rate and 95% CI |
| | | 62% (44%, 80%) Sensitivity and 95% CI | 99% (99%, 99%) Specificity and 95% CI | (n=1809) |
| | | Yes | No | |
| HAT from HAT RCA database | Yes | 7 | 0 | 100% (100%, 100%) True positive rate and 95% CI |
| | No | 19 | 1542 | 1% (1%, 2%) False negative rate and 95% CI |
| | | 27% (10%, 44%) Sensitivity and 95% CI | 100% (100%, 100%) Specificity and 95% CI | (n=1568)* |
| | | Major bleed from case note review | | |
| | | Yes | No | |
| Bleed from ICD-10/ SNOMED codes | Yes | 14 | 68 | 17% (9%, 25%) True positive rate and 95% CI |
| | No | 31 | 1696 | 2% (1%, 2%) False negative rate and 95% CI |
| | | 31% (18%, 45%) Sensitivity and 95% CI | 96% (95%, 97%) Specificity and 95% CI | (n=1809) |

n=1809 following removal of H/B/V batch patients.
*Manchester unable to access HAT database.
HAT, hospital-acquired thrombosis; ICD-10, International Classification of Diseases 10th Revision; RCA, route cause analysis.

between types of VTE. Our findings align with these latter reports but offer additional validation of HAT states (in addition to VTE diagnosis) compared with routine data.

Several authors have experimented with composite data sets and diagnostic/procedural/disease coding combinations, similar to our work. One study combined ICD-10 codes for VTE with a common procedural terminology code for a VTE Diagnostic Study plus at least one of the following within 30 days of diagnosis; pharmacy script for anticoagulation, placement of an inferior vena cava filter or death.[20] This algorithm still lacked sensitivity, reporting a value of 0.67 (0.60, 0.73) although corresponding specificity was high at 0.99 (0.98, 0.99). Alotaibi et al subsequently combined routinely collected ICD-10 coding data with imaging procedure codes to identify VTE events over a 10-year period, compared with case note review. Again, they report highly specific results but limited sensitivity, in line with our findings (74.83% (95% CI, 67.01 to 81.62) and 75.24% (95% CI, 65.86 to 83.14) for PE and DVT, respectively).[21] Verma et al report using natural language processing algorithms for digital interrogation of radiology reports in a large cohort of hospitalised medical patients to identify VTE outcomes.[19] The authors conclude a high level of accuracy, reporting sensitivities of 94%/91% and PPVs of 90%/89% for DVT and PE, respectively. Finally, Klil-Drori et al have recently

validated an algorithm for confirmation of suspected PE, combining emergency department diagnosis coding, imaging coding and dispensed prescription or hospital treatment.[22] The authors report overall agreement of their algorithm with confirmed PE (adjudicated through chart review) in 92.2% cases. Again, such an algorithm would not discriminate between index diagnosis of VTE and subsequent development of HAT. Such algorithms also require external validation in a UK setting.

In 2017, Baumgartner et al highlighted further issues through interrogation of an administrative coding database, looking to determine the accuracy of ICD-10 coding for new episodes of recurrent VTE in patients with a history.[23] Only 31.1% of coded encounters were verified by reviewers as true recurrent VTE. More recently, Pellathy et al have conducted similar work within the USA, comparing accuracy of HAT diagnoses made through administrative coding to manual case note and radiology review.[24] The authors report that only 40% of HAT cases identified through routine coding were confirmed by case note review and 45% of HAT confirmed through diagnostic test records lacked corresponding ICD codes.

### Meaning of the study
There are multiple potential explanations for the limited performance of routine data to identify HAT.

The condition is a temporal phenomenon and routine coding data can therefore mistake index presentation with VTE as HAT (false positive); patients who present with symptoms but wait >48 hours for radiological confirmation of diagnosis would erroneously fit the conventional definition of HAT (VTE occurring >24 hours from hospital admission). International guidelines also now support outpatient diagnosis and management of VTE, so genuine cases of HAT may not require hospital admission or receive appropriate coding (false negative). These two factors are the most important contributors to poor internal validity of efficient data methods, reflected in several studies across different countries.[19 23 24] In particular, Fang et al highlight the poor performance of outpatient coding to predict VTE in a separate cohort of 4642 adult patients.[25] Finally, coding teams may fail to document subsequent HAT (false negative) following index admission with alternative pathology (such as pneumonia) and prior diagnosis of VTE can often be coded during repeat hospital attendance, mistaken for HAT (false positive). In the case of major bleeding, we found that coding of disease states with potential for bleeding (but without actual bleeding) was the biggest contributing factor to the high rate of false positive results. This issue arose due to strict definitions of major bleeding as per ISTH definition which are not mirrored by an ICD coding structure.[10]

Most UK hospitals conducting RCA of HAT cases in line with NHS contract standards have developed pathways to mitigate these issues, through local reporting arrangements with radiology and pathology. Local leads extract all cases of DVT and PE identified by their Radiology and Ultrasound services and assess whether there was a hospital admission within 90 days prior to the VTE; if so they conduct RCA by reviewing the patients notes to assess whether the VTE was potentially preventable. Such arrangements often work well, but are reliant on individuals and reporting systems subject to human error. These issues are reflected in our findings, which report a PPV of 100% for HAT RCA database findings, but limited sensitivity (implying local identification of positive cases is accurate, but missed cases still occur despite a systematic approach).

### Possible explanations and implications for clinicians and policymakers

More generally, these findings raise questions about the current enthusiasm for data enabled trials when outcomes are complex.[26] Such concepts are inherently attractive to researchers and patients, particularly in topic areas with low event rates. However, complex outcome measures which require temporal evaluation and qualification against prior disease states are unlikely to be reliably delivered through the use of routinely collected data in isolation. For example, relevant data may contain coding errors arising from ambiguous documentation by physicians and inconsistent definitions.[27 28] Recent case studies have reported significant amounts of missing data

and poor interobserver agreement between routinely collected EHR data accessible through HES and case report form evaluation.[29] Electronic records contain an abundance of free text, but often lack necessary intelligence to classify patient episodes appropriately, or allow processing and comparison of routinely collected data.[30] Increasing complexity in outcome is also likely to correspond with decreasing accuracy of routine data. A registry study of Medicare claims following mitral valve repair compared with formal adjudication reported a PPV for mortality of 97%, heart failure requiring hospitalisation of 69%, bleeding of 40% and renal failure of 19%.[31]

In addition, the time and effort needed to acquire necessary permissions for national routine coding data or to orchestrate data linkage can be substantial. A UK clinical trials unit recently reported a digital request in the context of a randomised controlled trial, highlighting a negotiation process over consent that took several years. Even after consent, the study team were in receipt of data 15 months following application.[32] Such timeframes may only be realistic within the context of continually adaptive design trials.

### Unanswered questions and future research

This work is restricted primarily to medical, surgical and orthopaedic patients. We did not evaluate efficient data methods for VTE or bleeding events in specific patient subgroups, such as cancer or neurosurgery. In addition, our work is UK based; other countries may be able to demonstrate more confidence in the accuracy of routinely collected data, although our review of the literature does not support this theory.

In their call to action, Sydes et al discuss supplementation of trial specific follow-up as an option to realise the full potential of data-enabled research.[26] Such an approach has potential merit to attempt identification of potential HAT, given the high PPV and high specificity of routine data sources. In addition, routine data sources may have a role in other research contexts, such as identification of cases for qualitative work, case–control studies, targeted individual follow-up or downstream survey work.

### CONCLUSIONS

Our study highlights the potential limitations of using routine data methods in the context of future research on VTE risk assessment. Such methods identify both false negative and false positive VTE cases, through failure to identify ambulatory cases without formal hospital coding and overdiagnosis of prior disease. Our findings were similar with regard to bleeding events, showing poor sensitivity of ICD-10 coding data and multiple false positive events identified across four sites. These findings have implications for funders looking to support further work in this area and suggest that large studies reliant on routine data collection methods in isolation are likely to be inaccurate and therefore unfeasible.

**Author affiliations**
<sup>1</sup>Emergency Department, Northern Care Alliance NHS Foundation Trust, Salford, UK
<sup>2</sup>Division of Infection, Immunity and Respiratory Medicine, The University of Manchester, Manchester, UK
<sup>3</sup>School of Health and Related Research (ScHARR), The University of Sheffield, Sheffield, UK
<sup>4</sup>Division of Cardiovascular Sciences, The University of Manchester, Manchester, UK
<sup>5</sup>Department of Medicine, McMaster University, Hamilton, Ontario, Canada
<sup>6</sup>Emergency Department, Hamilton General Hospital, Hamilton, Ontario, Canada
<sup>7</sup>Medical Care Research Unit, University of Sheffield, Sheffield, UK
<sup>8</sup>Kings Healthcare Partners & Thrombosis & Haemophilia Centre, Guy's and St Thomas' NHS Foundation Trust, London, UK

**Presented at** Sections of this work were presented at the ISTH 2022 conference in London by poster. This work is also accepted for publication by the NIHR in monograph format within their journals library in April 2023. The NIHR strongly encourages wide dissemination of funded work and early submission of manuscripts, as per the embargo policy: 'In line with our embargo policy, we are prepared to delay publication of your Journals Library manuscript and, where possible, coordinate publication of your Journals Library manuscript with any journal articles'. More details on the NIHR Embargo policy can be found at the following address: https://www.journalslibrary.nihr.ac.uk/policies/#Embargo.

**Acknowledgements** We would like to acknowledge the support of the research nurses and assistants involved in chart review, data extraction and entry across the four hospital sites, including Reece Doonan, Efia Mainoo, Linda Debattista, Sarah Bird and Anna Wilson. We would also like to acknowledge the wider group directly conducting the VTEAM project (NIHR 127454), including project manager Helen Shulver, literature expert Abdullah Pandor, clerical assistant Heather Dakin, topic expert Xavier Griffin and clinical expert Mark Holland. We would also like to acknowledge the valuable input from the patient and public representatives, Robin Pierce-Williams, Chris Tweedy, Ben Langsdale, Deb Smith (Thrombosis UK), Shan Bennett and Enid Hirst (Sheffield Emergency Care Forum).

**Contributors** The authors were involved as follows: SG and DH (conception), CR, SG, BJH and DH (execution, analysis and drafting manuscript). SR designed and developed the iterative database. MBu and MBr conducted statistical evaluation of the data set on behalf of the CTRU. KdW and BJH attended Project Management Group meetings and contributed to drafting of the final manuscript. All authors were involved in critical discussion, revision and final approval of the manuscript. DH acts as guarantor. The corresponding author attests that all listed authors meet authorship criteria and that no others meeting the criteria have been omitted.

**Funding** This study was funded by the UK National Institute for Health Research (NIHR) Health Technology Assessment (HTA) Programme (project number 127454).

**Disclaimer** The views expressed in this report are those of the authors and not necessarily those of the NIHR HTA Programme. Any errors are the responsibility of the authors. The funders had no role in the study design, in the collection, analysis and interpretation of data; in the writing of the manuscript and in the decision to submit the manuscript for publication.

**Competing interests** During the completion of this study, SG, DH, CR, BJH, MBu and MB received funding from the National Institute of Health Research (NIHR) for academic work in this area, through competitive grant application and CR was appointed to an NIHR doctoral research fellow position. Following the completion of this study, CR has been subsequently employed by Pfizer limited. Pfizer did not fund nor support this study and was not involved in drafting or revising this manuscript.

**Patient and public involvement** Patients and/or the public were involved in the design, or conduct, or reporting or dissemination plans of this research. Refer to the Methods section for further details.

**Patient consent for publication** Not required.

**Ethics approval** This study involves human participants. The study received a favourable opinion from the Proportionate Review Subcommittee of the London—West London & GTAC Research Ethics Committee and approval from the HRA and Care Research Wales (HCRW) on 18 September 2019 (reference 19/LO/1303, IRAS project ID 262220). This work involved accessing routine healthcare data only and did not impact on clinical care or patient experience.

**Provenance and peer review** Not commissioned; externally peer reviewed.

**Data availability statement** Data are available upon reasonable request. Requests for data access should be made in writing to the School for Health and Related Research at the University of Sheffield.

**ORCID iDs**
Daniel Horner http://orcid.org/0000-0002-0400-2017
Charles Reynard http://orcid.org/0000-0002-7534-2668
Mike Bradburn http://orcid.org/0000-0002-3783-9761
Kerstin de Wit http://orcid.org/0000-0003-2763-6474
Steve Goodacre http://orcid.org/0000-0003-0803-8444

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
