## [Reviewer comments · BMJ Open]

ARTICLE DETAILS

TITLE (PROVISIONAL)	The accuracy of efficient data methods to determine the incidence of hospital acquired thrombosis and major bleeding in medical and surgical inpatients: a multicentre observational cohort study in four UK hospitals
AUTHORS	Horner, Daniel; Rex, Saleema; Reynard, Charles; Bursnall, Matthew; Bradburn, Mike; de Wit, Kerstin; Goodacre, Steve; Hunt, Beverley

VERSION 1 – REVIEW

REVIEWER	Okumura, Yasuo Division of Cardiology, Department of Medicine, Nihon University School of Medicine
REVIEW RETURNED	13-Nov-2022

GENERAL COMMENTS	Horner D et al. evaluated the accuracy of using routine health service data to identify hospital acquired thrombosis (HAT) and major bleeding events (MBE) compared to a reference standard of case note review. They found that using routine data instead of case note review would be highly likely to miss a substantial proportion of outcome events. Overall, their manuscript was well-written, and analyzed. I have some comments. 1. In the discussion section, they addressed multiple potential explanations for the limited performance of routine data to identify HAT. What is the most important factors for the limited performance of routine data? Do they have any supporting data for several potential explanations that they addressed? 2. What characteristics exist in VTE cases with false negative and false positive, and bleeding cases with a false positive? Can they expand results and discussion about this point?
--

REVIEWER	Zrelak, Patricia A Kaiser Permanente
REVIEW RETURNED	05-Dec-2022

GENERAL COMMENTS	Overall, this is a very well-written paper evaluating the accuracy of using routine data to identify Hospital Acquired Thrombosis (HAT) and related major bleeding events (MBE) compared to case note examination. Major comment and concern: Page 8, line 26. - Why did you include events typically not clinically significant, such as phlebitis and superficial vessels? This is a major limitation of the work that could have negatively impacted results. Also, the prevention potentially differs based on etiology. Likewise, I'm concerned that many of the bleeding events could
---

	have had other contributing causes other than anticoagulation. I would like to have seen this flushed out. It may be that the catchment is too large. Minor comments: Page 7. Line 32. How the A batches were handled needs more explanation. It is assumed you did this to make the work manageable. Page 7, line 58. Can you elaborate on the validity of the Department of Health tool to facilitate VTE risk assessment? Page 8, line 5. Can you further define shared primary and secondary care electronic health records? I assume you are referring to the patient's primary medical record and other databases without specification. Page 8, line 21. I think you mean you collected data elements from multiple data sources. Page 8, line 57. I'm uncertain how cases were identified for batch H - the HAT database entries. Figure 1 - site 2. I would recopy these pages to be more eligible (1 sheet flat on the scanner -and 1 page per picture). Figure 2 contains some undefined abbreviations, such as CTRU. Table 3- What is a HAT RCA database?
--	---

VERSION 1 – AUTHOR RESPONSE

Reviewer 1 comments:

2. In the discussion section, they addressed multiple potential explanations for the limited performance of routine data to identify HAT. What is the most important factors for the limited performance of routine data? Do they have any supporting data for several potential explanations that they addressed?

Thanks. We have revised this paragraph to highlight what we perceive to be the most important factors for poor performance of routine data sources. We have added references and cited several studies to support these statements, as advised.

3. What characteristics exist in VTE cases with false negative and false positive, and bleeding cases with a false positive? Can they expand results and discussion about this point?

We have expanded the relevant discussion paragraph ('meaning of the study') significantly to address these points. We have annotated the impact of specific issues/characteristics as resulting in false positive or false negatives respectively and added a sentence on the potential reasons for high false positive results in cases of major bleeding.

Reviewer 2:

4. Page 8, line 26. - Why did you include events typically not clinically significant, such as phlebitis and superficial vessels?

Thanks for the opportunity to clarify this. We included all ICD10 codes that we felt could be relevant to a diagnosis of deep vein thrombosis. Interestingly, all ICD10 codes are categorised as 'phlebitis or thrombophlebitis' initially, even when affecting the femoral vein (ICD10-180.1)! Given previous literature highlighted concerns around the sensitivity of routine data coding for detection of VTE, we

thought it preferable to cast the data net as wide as possible. We did not classify superficial venous thrombosis within our case definition of HAT, therefore at worst this issue would have only cause more false positive cases. It would not have affected sensitivity.

We have added a paragraph in the methods on our case definition of VTE to clarify the above issues for readers.

5. I'm concerned that many of the bleeding events could have had other contributing causes other than anticoagulation.

We agree. However, the aim of this study was to correlate disease states identified at case note review with disease states identified through efficient data methods, rather than explore individual variables associated with disease states. We have added sentences to the discussion and limitations to highlight this.

6 Page 7. Line 32. How the A batches were handled needs more explanation. It is assumed you did this to make the work manageable.

Yes, this was to deliver the iterative nature of the trial as effectively and efficiently as possible. We have expanded this section in the methods as advised.

7. 'Page 7, line 58. Can you elaborate on the validity of the Department of Health tool to facilitate VTE risk assessment?

We have now expanded on the structure, validity and use of this RAM over several sentences in the relevant section, with additional references.

8. Page 8, line 5. Can you further define shared primary and secondary care electronic health records? I assume you are referring to the patient's primary medical record and other databases without specification

This is correct, but thanks for highlighting. We have expanded this section to provide more clarity on the use of EHR and variation across site and care interface.

9. Page 8, line 21. I think you mean you collected data elements from multiple data sources.

Thanks, we have amended as suggested.

10. Page 8, line 57. I'm uncertain how cases were identified for batch H - the HAT database entries.

Noting point 13 below also, we have expanded our paragraph on HAT databases to provide more clarity on structure, co-ordination and utility. The relevant revised paragraph is highlighted below:

HAT database entries are co-ordinated by local thrombosis committees in accordance with NHS contract standards and include a contemporary register of all patients diagnosed with acute VTE at the hospital site, informed by radiology or identified by pathology at post mortem. All cases are subsequently categorised by the local thrombosis committee as either de-novo VTE or HAT based on case review, local expert opinion and data on any preceding hospital admission (up to 90 days) or VTE diagnosis >24h following hospital admission. This database is maintained contemporaneously and provides an ongoing opportunity for hospitals to identify preventable HAT and conduct root cause analysis (RCA) for each episode, to promote learning and best practice.

In addition, we have expanded the sentence on batches to clarify how cases were identified and

subsequently used within the wider dataset.

11. Figure 1 - site 2. I would recopy these pages to be more eligible (1 sheet flat on the scanner -and 1 page per picture).

We would be happy to provide better images in any format advised by the editorial team if the manuscript is accepted for submission.

12. Figure 2 contains some undefined abbreviations, such as CTRU.

Thanks. We have corrected.

13. Table 3- What is a HAT RCA database?

Please see our response to point 10 above.

VERSION 2 – REVIEW

REVIEWER	Okumura, Yasuo Division of Cardiology, Department of Medicine, Nihon University School of Medicine
REVIEW RETURNED	08-Jan-2023

GENERAL COMMENTS	No comments. They clearly addressed my requests.
--

REVIEWER	Zrelak, Patricia A Kaiser Permanente
REVIEW RETURNED	3-Jan-2023

GENERAL COMMENTS	There are a few minor wordsmithing suggestions that are not related to the scientific integrity of the article. 1. Recommend some wordsmithing to the competing interest statement as follows: On behalf of all authors, I declare the following competing interests: During the completion of this study, SG, DH, CR, BH, MBu and MB received funding from the National Institute of Health Research for academic work in this area, through a competitive grant application and appointment to a doctoral research fellow position (CR DEFINE CR). Following the completion of the study, CR has been employed by Pfizer limited. Pfizer did not fund nor support this study and was not involved in drafting or revising this manuscript. 2. Under the results section of the abstract. The first sentence is confusing. Did you abstract 87% of all eligible cases OR did you collect necessary routine data from 87% of the cases examined. It reads as the latter. Please confirm if correct. We were able to collect the necessary routine data from 87% of 2008 case episodes examined. 3. Considering defining NHS for the non-Brits. 4. Study population consider adding the work “records”.
---

	We collated cases (or records) in batches of 50 to facilitate iterative and incremental case ascertainment; initial hospital downloads were often in excess of 50,000 case episodes. In order to keep workflow manageable and organized, we worked through batches of 50 records and reported routinely to a steering committee, who provided guidance on study delivery. We collated data on all risk assessments that clinical staff performed prospectively at the point of hospital admission, then scrambled episodes into randomly assorted batches of 50 (A batches) to ensure diversity in specialty presentation and mitigate seasonal bias. Suggested change: We collated data on all risk assessments that clinical staff performed prospectively at the point of hospital admission, then scrambled episodes into randomly assorted batches of 50 (referred to as A batches) to ensure diversity in specialty presentation and mitigate seasonal bias. Define CTPA. (page 16)
--	--

VERSION 2 – AUTHOR RESPONSE

viewer 1 comments:

1. Recommend some wordsmithing to the competing interest statement as follows:

Thanks - we are grateful for direct suggestions and have revised appropriately in keeping with all minor recommendations here.

2. Under the results section of the abstract. The first sentence is confusing.

We have revised this sentence for clarity.

3. Considering defining NHS for the non-Brits.

Done.

4. Study population consider adding the work “records”

Thanks - we are grateful for direct suggestions and have revised appropriately in keeping with all minor recommendations here.

5. Define CTPA. (page 16)

Done.